# BALANCING CONSTRAINTS AND REWARDS WITH META-GRADIENT D4PG

**Dan A. Calian\*, Daniel J. Mankowitz\*, Tom Zahavy, Zhongwen Xu,**
**Junhyuk Oh, Nir Levine & Timothy Mann**
DeepMind
London, United Kingdom
`{dancalian, dmankowitz}@google.com`

## ABSTRACT

Deploying Reinforcement Learning (RL) agents to solve real-world applications often requires satisfying complex system constraints. Often the constraint thresholds are incorrectly set due to the complex nature of a system or the inability to verify the thresholds offline (e.g, no simulator or reasonable offline evaluation procedure exists). This results in solutions where a task cannot be solved without violating the constraints. However, in many real-world cases, constraint violations are undesirable yet they are not catastrophic, motivating the need for soft-constrained RL approaches. We present a soft-constrained RL approach that utilizes meta-gradients to find a good trade-off between expected return and minimizing constraint violations. We demonstrate the effectiveness of this approach by showing that it consistently outperforms the baselines across four different Mujoco domains.

## 1 INTRODUCTION

Reinforcement Learning (RL) algorithms typically try to maximize an expected return objective (Sutton & Barto, 2018). This approach has led to numerous successes in a variety of domains which include board-games (Silver et al., 2017), computer games (Mnih et al., 2015; Tessler et al., 2017) and robotics (Abdolmaleki et al., 2018). However, formulating real-world problems with only an expected return objective is often sub-optimal when tackling many applied problems ranging from recommendation systems to physical control systems which may include robots, self-driving cars and even aerospace technologies. In many of these domains there are a variety of challenges preventing RL from being utilized as the algorithmic solution framework. Recently, Dulac-Arnold et al. (2019) presented nine challenges that need to be solved to enable RL algorithms to be utilized in real-world products and systems. One of those challenges is handling constraints. All of the above domains may include one or more constraints related to cost, wear-and-tear, or safety, to name a few.

**Hard and Soft Constraints:** There are two types of constraints that are encountered in constrained optimization problems; namely hard-constraints and soft-constraints (Boyd & Vandenberghe, 2004). Hard constraints are pairs of pre-specified functions and thresholds that require the functions, when evaluated on the solution, to respect the thresholds. As such, these constraints may limit the feasible solution set. Soft constraints are similar to hard constraints in the sense that they are defined by pairs of pre-specified functions and thresholds, however, a soft constraint does not require the solution to hold the constraint; instead, it penalizes the objective function (according to a specified rule) if the solution violates the constraint (Boyd & Vandenberghe, 2004; Thomas et al, 2017).

**Motivating Soft-Constraints:** In real-world products and systems, there are many examples of soft-constraints; that is, constraints that can be violated, where the violated behaviour is undesirable but not catastrophic (Thomas et al., 2017; Dulac-Arnold et al., 2020b). One concrete example is that of energy minimization in physical control systems. Here, the system may wish to reduce the amount of energy used by setting a soft-constraint. Violating the constraint is inefficient, but not catastrophic to the system completing the task. In fact, there may be desirable characteristics that can only be attained if there are some constraint violations (e.g., a smoother/faster control policy). Another common setting is where it is unclear how to set a threshold. In many instances, a product

---

\* indicates equal contribution.

manager may desire to increase the level of performance on a particular product metric $A$, while ensuring that another metric $B$ on the same product does not drop by 'approximately X%'. The value 'X' is often inaccurate and may not be feasible in many cases. In both of these settings, violating the threshold is undesirable, yet does not have catastrophic consequences.

**Lagrange Optimization:** In the RL paradigm, a number of approaches have been developed to incorporate hard constraints into the overall problem formulation (Altman, 1999; Tessler et al., 2018; Efroni et al., 2020; Achiam et al., 2017; Bohez et al., 2019; Chow et al., 2018; Paternain et al., 2019; Zhang et al., 2020; Efroni et al., 2020). One popular approach is to model the problem as a Constrained Markov Decision Process (CMDP) (Altman, 1999). In this case, one method is to solve the following problem formulation: $\max_\pi J_R^\pi$ s.t. $J_C^\pi \leq \beta$, where $\pi$ is a policy, $J_R^\pi$ is the expected return, $J_C^\pi$ is the expected cost and $\beta$ is a constraint violation threshold. This is often solved by performing alternating optimization on the unconstrained Lagrangian relaxation of the original problem (e.g. Tessler et al. (2018)), defined as: $\min_{\lambda \geq 0} \max_\pi J_R^\pi + \lambda(\beta - J_C^\pi)$. The updates alternate between learning the policy and the Lagrange multiplier $\lambda$.

In many previous constrained RL works (Achiam et al., 2017; Tessler et al., 2018; Ray et al., 2019; Satija et al., 2020), because the problem is formulated with hard constraints, there are some domains in each case where a feasible solution is not found. This could be due to approximation errors, noise, or the constraints themselves being infeasible. The real-world applications, along with empirical constrained RL research results, further motivates the need to develop a soft-constrained RL optimization approach. Ideally, in this setup, we would like an algorithm that satisfies the constraints while solving the task by maximizing the objective. If the constraints cannot be satisfied, then this algorithm finds a good trade-off (that is, minimizing constraint violations while solving the task by maximizing the objective).

In this paper, we extend the constrained RL Lagrange formulation to perform soft-constrained optimization by formulating the constrained RL objective as a nested optimization problem (Sinha et al., 2017) using meta-gradients. We propose MetaL that utilizes meta-gradients (Xu et al., 2018; Zahavy et al., 2020) to improve upon the trade-off between reducing constraint violations and improving expected return. We focus on Distributed Distributional Deterministic Policy Gradients (D4PG) (Barth-Maron et al., 2018) as the underlying algorithmic framework, a state-of-the-art continuous control RL algorithm. We show that MetaL can capture an improved trade-off between expected return and constraint violations compared to the baseline approaches. We also introduce a second approach called MeSh that utilizes meta-gradients by adding additional representation power to the reward shaping function. Our **main contributions** are as follows: **(1)** We extend D4PG to handle constraints by adapting it to Reward Constrained Policy Optimization (RCPO) (Tessler et al., 2018) yielding Reward Constrained D4PG (RC-D4PG); **(2)** We present a soft constrained meta-gradient technique: **Meta**-Gradients for the **L**agrange multiplier learning rate (MetaL)[1]; **(3)** We derive the meta-gradient update for MetaL (Theorem 1); **(4)** We perform extensive experiments and investigative studies to showcase the properties of this algorithm. MetaL outperforms the baseline algorithms across domains, safety coefficients and thresholds from the Real World RL suite (Dulac-Arnold et al., 2020b).

## 2 BACKGROUND

A **Constrained Markov Decision Process (CMDP)** is an extension to an MDP (Sutton & Barto, 2018) and consists of the tuple $\langle S, A, P, R, C, \gamma \rangle$ where $S$ is the state space; $A$ is the action space; $P : S \times A \to S$ is a function mapping states and actions to a distribution over next states; $R : S \times A \to \mathbb{R}$ is a bounded reward function and $C : S \times A \to \mathbb{R}^K$ is a $K$ dimensional function representing immediate penalties (or costs) relating to $K$ constraints. The solution to a CMDP is a policy $\pi : S \to \Delta_A$ which is a mapping from states to a probability distribution over actions. This policy aims to maximize the expected return $J_R^\pi = \mathbb{E}[\sum_{t=0}^\infty \gamma^t r_t]$ and satisfy the constraints $J_{C_i}^\pi = \mathbb{E}[\sum_{t=0}^\infty \gamma^t c_{i,t}] \leq \beta_i, i = 1 \dots K$. For the purpose of the paper, we consider a single constraint; that is, $K = 1$, but this can easily be extended to multiple constraints.

**Meta-Gradients** is an approach to optimizing hyperparameters such as the discount factor, learning rates, etc. by performing online cross validation while simultaneously optimizing for the overall RL optimization objective such as the expected return (Xu et al., 2018; Zahavy et al., 2020). The goal is to optimize both an inner loss and an outer loss. The update of the $\theta$ parameters on the *inner*

---

[1]This is also the first time meta-gradients have been applied to an algorithm with an experience replay.

*loss* is defined as $\theta' = \theta + f(\tau, \theta, \eta)$, where $\theta \in \mathbb{R}^d$ corresponds to the parameters of the policy $\pi_\theta(a|s)$ and the value function $v_\theta(s)$ (if applicable). The function $f : \mathbb{R}^k \to \mathbb{R}^d$ is the gradient of the policy and/or value function with respect to the parameters $\theta$ and is a function of an n-step trajectory $\tau = \langle s_1, a_1, r_2, s_2 \dots s_n \rangle$, meta-parameters $\eta$ and is weighted by a learning rate $\alpha$ and is defined as $f(\tau, \theta, \eta) = \alpha \frac{\mathrm{d}J_{obj}^{\pi_\theta}(\theta, \tau, \eta)}{\mathrm{d}\theta}$ where $J_{obj}^{\pi_\theta}(\theta, \tau, \eta)$ is the objective being optimized with respect to $\theta$. The idea is to then evaluate the performance of this new parameter value $\theta'$ on an *outer loss* – the meta-gradient objective. We define this objective as $J'(\tau', \theta', \bar{\eta})$ where $\tau'$ is a new trajectory, $\theta'$ are the updated parameters and $\bar{\eta}$ is a *fixed* meta-parameter (which needs to be selected/tuned in practice). We then need to take the gradient of the objective $J'$ with respect to the meta-parameters $\eta$ to yield the outer loss update $\eta' = \eta + \alpha_\eta \frac{\partial J'(\tau', \theta', \bar{\eta})}{\partial \eta}$. This gradient is computed as follows: $\frac{\partial J'(\tau', \theta', \bar{\eta})}{\partial \eta} = \frac{\partial J'(\tau', \theta', \bar{\eta})}{\partial \theta'} \frac{\partial \theta'}{\partial \eta}$. The outer loss is essentially the objective we are trying to optimize. This could be a policy gradient loss, a temporal difference loss, a combination of the two etc (Xu et al., 2018; Zahavy et al., 2020). Meta-gradients have been previously used to learn intrinsic rewards for policy gradient (Zheng et al., 2018) and auxiliary tasks (Veeriah et al., 2019). Meta-gradients have also been used to adapt optimizer parameters (Young et al., 2018; Franceschi et al., 2017). In our setup, we consider the continuous control setting, provide the first implementation of meta-gradients for an algorithm that uses an experience replay, and focus on adapting meta-parameters that encourage soft constraint satisfaction while maximizing expected return.

**D4PG** is a state-of-the-art continuous control RL algorithm with a deterministic policy (Barth-Maron et al., 2018). It is an incremental improvement to DDPG (Lillicrap et al., 2015). The overall objective of DDPG is to maximize $J(\theta_a, \theta_c) = \mathbb{E}[Q_{\theta_c}(s, a)|s = s_t, a = \pi_{\theta_a}(s_t)]$ where $\pi_{\theta_a}(s_t)$ is a deterministic policy with parameters $\theta_a$ and $Q_{\theta_c}(s, a)$ is an action value function with parameters $\theta_c$. The actor loss is defined as: $L_{actor} = \|\text{SG}(\nabla_a Q_{\theta_c}(s_t, a_t)|_{a_t = \pi_{\theta_a}(s)} + a_{\theta_a, t}) - a_{\theta_a, t}\|_2$ where SG is a stop gradient. The corresponding gradient update is defined as $\nabla_{\theta_a} J(\theta_a) = \mathbb{E}[\nabla_a Q_{\theta_c}(s, a) \nabla_{\theta_a} \pi_{\theta_a}(s_t)]$. The critic is updated using the standard temporal difference error loss: $L_{critic} = (r(s, a) + \gamma Q_T(s', \pi_T(s')) - Q_{\theta_c}(s, a))^2$ where $Q_T, \pi_T$ are the target critic and actor networks respectively. In D4PG, the critic is a distributional critic based on the C51 algorithm (Bellemare et al., 2017) and the agent is run in a distributed setup with multiple actors executed in parallel, n-step returns and with prioritized experience replay. We will use the non-distributional critic update in our notation for ease of visualization and clarity for the reader[2].

## 3  REWARD CONSTRAINED D4PG (RC-D4PG)

This section describes our modifications required to transform D4PG into Reward Constrained D4PG (RC-D4PG) such that it maximizes the expected return and satisfies constraints.

The constrained optimisation objective is defined as: $\max_{\pi_\theta} J_R^{\pi_\theta}$ subject to $J_C^{\pi_\theta} \leq \beta$, where $J_R^{\pi_\theta} = \mathbb{E}[Q(s, a)|s = s_t, a = \pi_\theta(s_t)]$ and $J_C^{\pi_\theta} = \mathbb{E}[C(s, a)|s = s_t, a = \pi_\theta(s_t)]$; the parameter $\theta = \langle \theta_a, \theta_c \rangle$ from here on in; $C(s, a)$ is a long-term penalty value function (e.g., sum of discounted immediate penalties) corresponding to constraint violations. The Lagrangian relaxation objective is defined as $J_R^{\pi_\theta} + \lambda(\beta - J_C^{\pi_\theta})$. As in RCPO, a proxy objective $J_R^{\pi_\theta} - \lambda J_C^{\pi_\theta}$ is used that converges to the same set of locally optimal solutions as the relaxed objective (Tessler et al., 2018). Note that the constant $\beta$ does not affect the policy improvement step and is only used for the Lagrange multiplier loss update. To optimize the proxy objective with D4PG, reward shaping of the form $r(s, a) - \lambda c(s, a)$ is required to yield the reward shaped critic loss defined as: $L_{critic}(\theta_c, \lambda) = (r(s, a) - \lambda c(s, a) + \gamma Q_T(s, \pi_T(s')) - Q_{\theta_c}(s, a))^2$. The actor loss is defined as before. The Lagrange loss is defined as: $L_{lagrange}(\lambda) = \lambda(\beta - J_C^{\pi_\theta})$ where $\lambda \geq 0$. Since RC-D4PG is off-policy, it requires storing the per time-step penalties, $c$, inside the transitions stored in the experience replay buffer (ER). For training the Lagrange multiplier an additional penalty buffer is used to store the per-episode penalties $J_C^{\pi_\theta}$. The learner then reads from this penalty buffer for updating the Lagrange multiplier. RC-D4PG updates the actor/critic parameters and the Lagrange multipliers using alternating optimization. The full algorithm for this setup can be found in the Appendix, Algorithm 3.

---

[2]This can easily be extended to include the distributional critic.

## 4 META-GRADIENTS FOR THE LAGRANGE LEARNING RATE (METAL)

In this section, we introduce the MetaL algorithm which extends RC-D4PG to use meta-gradients for adapting the learning rate of the Lagrangian multiplier[3]. The idea is to update the learning rate such that the outer loss (as defined in the next subsection) is minimized. Our intuition is that a learning rate gradient that takes into account the overall task objective and constraint thresholds will lead to improved overall performance.

---

**Algorithm 1** MetaL

1: **Input:** penalty $c(\cdot)$, constraint $C(\cdot)$, threshold $\beta$, learning rates $\alpha_1, \alpha_{\theta_a}, \alpha_{\theta_c}, \alpha_\eta$, max. number of episodes $M$
2: Initialize actor, critic parameters $\theta_a$ and $\theta_c$, Lagrange multiplier $\lambda = 0$, meta-parameter $\eta = \alpha_\lambda$
3: **for** $1 \ldots M$ **do**
4:     **Inner loss:**
5:     Sample episode penalty $J_C^\pi$ from the penalty replay buffer
6:     $\lambda' \leftarrow [\lambda - \alpha_1 \exp(\alpha_\lambda)(\beta - J_C^\pi)]_+$           ▷ **Lagrange multiplier update**
7:     Sample a batch with tuples $\langle s_t, a_t, r_t, c_t \rangle_{t=1}^T$ from ER and split into training/validation sets
8:     Accumulate and apply actor and critic updates over training batch $T_{train}$ by:
9:     $\nabla \theta_c = 0, \nabla \theta_a = 0$
10:     **for** $t = 1 \ldots T_{train}$ **do**
11:         $\hat{R}_t = r_t - \lambda' c_t + \gamma \hat{Q}(\lambda, s_{t+1}, a_{t+1} \sim \pi_T(s_{t+1}); \theta_c)$
12:         $\nabla \theta_c \mathrel{+}= \alpha_{\theta_c} \partial(\hat{R}_t - \hat{Q}(\lambda, s_t, a_t; \theta_c))^2 / \partial \theta_c$     ▷ **Critic update**
13:         $\nabla \theta_a \mathrel{+}= \alpha_{\theta_a} \mathbb{E}[\nabla_a Q(s_t, a_t) \nabla_{\theta_a} \pi_{\theta_a}(s_t)|_{a_t = \pi(s_t)}]$    ▷ **Actor update**
14:     $\theta'_c \leftarrow \theta_c - \frac{1}{T_{train}} \sum \nabla \theta_c$
15:     $\theta'_a \leftarrow \theta_a + \frac{1}{T_{train}} \sum \nabla \theta_a$
16:     **Outer loss:** Compute outer loss and meta-gradient update using validation set $T_{validate}$:
17:     $\alpha'_\lambda \leftarrow \alpha_\lambda - \alpha_\eta \frac{\partial J'(\theta'_c(\alpha_\lambda), \lambda'(\alpha_\lambda))}{\partial \alpha_\lambda}$     ▷ **Meta-parameter update - Theorem 1**
18:     $\lambda \leftarrow \lambda', \theta_a \leftarrow \theta'_a, \theta_c \leftarrow \theta'_c, \alpha_\lambda \leftarrow \alpha'_\lambda$
19: **return** $\theta_a, \theta_c, \lambda$

---

**Meta-parameters, inner and outer losses:** The meta-parameter is defined as $\eta = \alpha_\lambda$. The *inner loss* is composed of three losses, the actor, critic and Lagrange loss respectively. The actor and critic losses are the same as in RC-D4PG. The Lagrange multiplier loss is defined as: $L_{lagrange}(\lambda) = \exp(\alpha_\lambda)\lambda(\beta - J_C^\pi)$ where $\alpha_\lambda$ is the meta-parameter as defined above. The meta-parameter is wrapped inside an exponential function to magnify the effect of $\alpha_\lambda$ while also ensuring non-negativity of the effective learning rate. The inner loss updates are

$$\begin{bmatrix} \theta'_a \\ \theta'_c \\ \lambda' \end{bmatrix} = \begin{bmatrix} \theta_a \\ \theta_c \\ \lambda \end{bmatrix} - \begin{bmatrix} f(\tau, \theta_a, \eta) \\ f(\tau, \theta_c, \eta) \\ f(\tau, \lambda, \eta) \end{bmatrix} = \begin{bmatrix} \theta_a \\ \theta_c \\ \lambda \end{bmatrix} - \begin{bmatrix} \alpha_{\theta_a} \frac{dL_{actor}(\theta_a)}{d\theta_a} \\ \alpha_{\theta_c} \frac{dL_{critic}(\theta_c, \lambda)}{d\theta_c} \\ \alpha_1 \frac{dL_{lagrange}(\lambda)}{d\lambda} \end{bmatrix}$$ where $\alpha_{\theta_a}, \alpha_{\theta_c}, \alpha_1$ are the fixed ac-

tor critic and Lagrange multiplier learning rates respectively. The *outer loss* is defined as $J'(\theta'_c(\alpha_\lambda), \lambda'(\alpha_\lambda)) = L_{outer} = L_{critic}(\theta'_c(\alpha_\lambda), \lambda'(\alpha_\lambda))$. We tried different variants of outer losses and found that this loss empirically yielded the best performance; we discuss this in more detail in the experiments section. This is analogous to formulating MetaL as the following nested optimization problem: $\min_{\alpha_\lambda} J'(\theta(\alpha_\lambda), \lambda(\alpha_\lambda))$, s.t. $\theta, \lambda \in \arg\min_{\theta, \lambda \geq 0} \{-J_R^{\pi_\theta} - \lambda(\alpha_\lambda)(\beta - J_C^{\pi_\theta})\}$. We treat the lower level optimization problem as the Lagrange relaxation objective (inner loss). We then treat the upper level optimization as the meta-gradient objective $J'(\theta(\alpha_\lambda), \lambda(\alpha_\lambda))$ (outer loss). This transforms the optimization problem into soft-constrained optimization since the meta-parameter $\alpha_\lambda$ guides the learning of the Lagrange multiplier $\lambda$ to minimize the outer loss while attempting to find a good trade-off between minimizing constraint violations and maximizing return (inner loss).

As shown in Algorithm 1, the inner loss gradients are computed for $\lambda$ (line 6), $\theta_c$ (line 12) and $\theta_a$ (line 13) corresponding to the Lagrange multiplier, critic and actor parameters respectively. The Lagrange multiplier is updated by sampling episode penalties which is an empirical estimate of $J_C^\pi$ from a separate penalty replay buffer (line 5) to compute the gradient update. The updated multiplier

---

[3]We plan on releasing the source code for MetaL in the near future.

is then utilized in the critic inner update (lines 11 and 12) to ensure that the critic parameters are a function of this new updated Lagrange multiplier. The actor and critic parameters are updated using the training batch, and these updated parameters along with a validation batch are used to compute the outer loss (line 17). The meta-parameter $\alpha_\lambda$ is then updated along the gradient of this outer loss with respect to $\eta = \alpha_\lambda$. We next derive the meta-gradient update for $\alpha_\lambda$, and present it in the following theorem (see the Appendix, Section A for the full derivation). Intuition for this meta-gradient update is provided in the experiments section.

**Theorem 1.** *MetaL gradient update: Let $\beta \geq 0$ be a pre-defined constraint violation threshold, meta-parameter $\eta = \alpha_\lambda$ and $J_C^{\pi_\theta} = \mathbb{E}\left[C(s,a)|s=s_t, a=\pi_\theta(s_t)\right]$ is the discounted constraint violation function, then, the meta-gradient update is:*

$$\alpha_\lambda' \leftarrow \alpha_\lambda - \alpha_\eta \left( -2\delta \cdot c(s,a) \cdot \alpha_1 \exp(\alpha_\lambda) \cdot \left( J_C^{\pi_\theta} - \beta \right) \left( -2\alpha_{\theta_c} (\nabla_{\theta_c'} Q_{\theta_c'}(s,a))^T \nabla_{\theta_c} Q_{\theta_c}(s,a) + 1 \right) \right) ,$$

*where $\delta$ is the TD error; $\alpha_{\theta_c}$ is the critic learning rate and $\alpha_\eta$ is the meta-parameter learning rate.*

## 5 EXPERIMENTS

The experiments were performed using domains from the Real-World Reinforcement Learning (RWRL) suite[4], namely cartpole:swingup, walker:walk, quadruped:walk and humanoid:walk. We will refer to these domains as cartpole, walker, quadruped and humanoid from here on in.

We focus on two types of tasks with constraints: (1) solvable constraint tasks - where the task is solved and the constraints can be satisfied; (2) unsolvable constraint tasks - where the task can be solved but the constraints *cannot* be satisfied. Unsolvable constraint tasks correspond to tasks where the constraint thresholds are incorrectly set and cannot be satisfied, situations which occur in many real-world problems as motivated in the introduction. The specific constraints we focused on for each domain can be found in the Appendix (Section C). The goal is to showcase the soft-constrained performance of MetaL, with respect to reducing constraint violations and maximizing the return in both of these scenarios (solvable and unsolvable constraint tasks) with respect to the baselines.

The baseline algorithms we focused on for each experiment are D4PG without any constraints, RC-D4PG (i.e., hard constraint satisfaction) and Reward Shaping D4PG (RS-D4PG) (i.e., soft constraint satisfaction). RS-D4PG uses a fixed $\lambda$ for the duration of training. We compare these baselines to MetaL. Note that D4PG, RC-D4PG and MetaL have *no prior information* regarding the Lagrange multiplier. RC-D4PG and MetaL attempt to learn a suitable multiplier value from scratch, i.e. the initial Lagrange multiplier value is set to 0.0. In contrast, RS-D4PG has prior information (i.e. it uses a pre-selected fixed Lagrange multiplier).

**Experimental Setup**: For each domain, the action and observation dimensions are shown in the Appendix, Table 4. The episode length is 1000 steps, the base reward function is computed within the dm_control suite (Tassa et al., 2018). The upper bound reward for each task is 1000. Each task was trained for 20000 episodes. Each variant of D4PG uses the same network architecture (see the Appendix, Table 5 for more details).

We use different performance metrics to compare overall performance. We track the average episode return ($R$), but we also define the **penalized return**: $R_{penalized} = R - \kappa \cdot \psi_{\beta,C}$, which captures the trade-off between achieving optimal performance and satisfying the constraints. Here, $R$ is the average return for the algorithm upon convergence (computed as an average over the previous 100 episodes); $\kappa$ is a fixed constant that determines how much to weight the constraint violation penalty. For the purposes of evaluation, we want to penalize algorithms that consistently violate the constraints and therefore set $\kappa = 1000$. Since the upper bound of rewards for each domain is 1000, we are essentially weighing equally attaining high performance and satisfying constraints. Finally, $\psi_{\beta,C} = \max(0, J_C^\pi - \beta)$ is defined as the **overshoot**. Here $\beta$ is the **constraint violation threshold** and defines the allowable average constraint violations per episode; $J_C^\pi$ is the average constraint violation value per episode upon convergence for a policy $\pi$. The overshoot, $\psi_{\beta,C}$, tracks the average constraint violations that are above the allowed constraint violation threshold $\beta$.

We investigate each algorithm's performance along a variety of dimensions which include different constraint violation thresholds (see the Appendix, Table 3 for a list of thresholds used), safety

---

[4]https://github.com/google-research/realworldrl_suite

coefficients and domains. The *safety coefficient* is a flag in the RWRL suite (Dulac-Arnold et al., 2020a). This flag contains values between $0.0$ and $1.0$. Reducing the value of the flag ensures that more constraint violations occur per domain per episode. As such, we searched over the values $\{0.05, 0.1, 0.2, 0.3\}$. These values vary from solvable constraint tasks (e.g., $0.3$) to unsolvable constraint tasks (e.g., $0.05$). We wanted to see how the algorithms behaved in these extreme scenarios. In addition, we analysed the performance across a variety of different constraint violation thresholds (see Appendix, Table 6). All experiments are averaged across 8 seeds.

## 5.1 MAIN RESULTS

We begin by analyzing the performance of our best variant, MetaL, with different outer losses. Then we analyse the overall performance of all methods, followed by dissecting performance along the dimensions of safety coefficient and domain respectively. Finally, we investigate the derived gradient update for MetaL from Theorem 1 and provide intuition for the algorithm's behaviour.

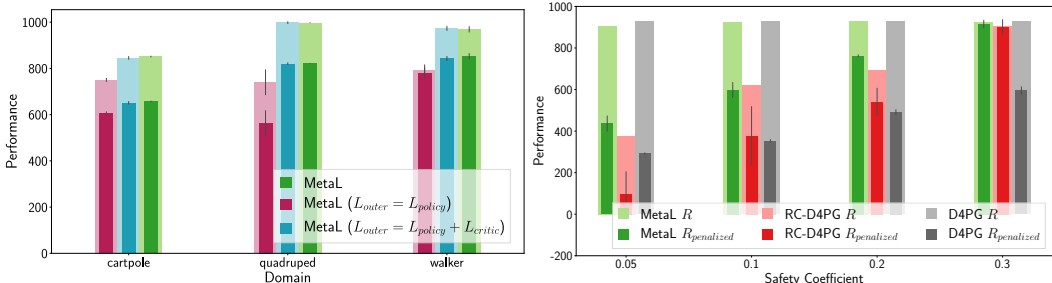

Figure 1: MetaL performance using different outer losses (left) and comparison with D4PG and RC-D4PG (right).

**MetaL outer loss**: We wanted to determine whether different outer losses would result in improved overall performance. We used the actor loss ($L_{actor}$) and the combination of the actor and critic losses as the outer loss ($L_{actor} + L_{critic}$) and compared them with the original MetaL outer loss ($L_{critic}$) as well as the other baselines. Figure 1 shows that using just the actor loss results in the worst performance; while using the critic loss always results in better performance. The best performance is achieved by the original critic-only MetaL outer loss.

There is some intuition for choosing a critic-only outer loss. In MetaL, the critic loss is a function of lambda. As a result, the value of lambda affects the agents ability to minimize this loss and therefore learn an accurate value function. In D4PG, an accurate value function (i.e., the critic) is crucial for learning a good policy (i.e., the actor). This is because the policy relies on an accurate estimate of the value function to learn good actions that maximize the return (see D4PG actor loss). This would explain why adding the actor loss to the outer loss does not have much effect on the final quality of the solution. However, removing the critic loss has a significant effect on the overall solution.

**Overall performance**: We averaged the performance of MetaL across all safety coefficients, thresholds and domains and compared this with the relevant baselines. As seen in Table 1, MetaL outperforms all of the baseline approaches by achieving the best trade-off of minimizing constraint violations and maximizing return[5]. This includes all of the soft constrained optimization baselines (i.e., RS-D4PG variants), D4PG as well as the hard-constrained optimization algorithm RC-D4PG. It is interesting to analyze this table to see that the best reward shaping variants are (1) $RS - 0.1$ which achieves comparable return, but higher overshoot and therefore lower penalized return; (2) $RS - 1.0$ which attains significantly lower return but lower overshoot resulting in lower penalized return. D4PG has the highest return, but this results in significantly higher overshoot. While RC-D4PG attains lower overshoot, it also yields significantly lower overall return. We now investigate this performance in more detail by looking at the performance per safety coefficient and per domain.

**Performance as a function of safety coefficient:** We analyzed the average performance per safety coefficient, while averaging across all domains and thresholds. As seen in Figure 1 (right), MetaL achieves comparable average return to that of D4PG. In addition, it significantly outperforms both

---

[5]MetaL's penalized reward ($R_{penalized}$) performance is significantly better than the baselines with all p-values smaller than $10^{-9}$ using Welch's t-test.

| Algorithm | $R_{penalized}$ | $R$ | $\max(0, J_C^\pi - \beta)$ |
|---|---|---|---|
| D4PG | $432.70 \pm 11.99$ | **927.66** | 0.49 |
| **MetaL** | **$677.93 \pm 25.78$** | 921.16 | 0.24 |
| RC-D4PG | $478.60 \pm 89.26$ | 648.42 | **0.17** |
| RS-0.1 | $641.41 \pm 26.67$ | 906.76 | 0.27 |
| RS-1.0 | $511.70 \pm 15.50$ | 684.30 | 0.17 |
| RS-10.0 | $208.57 \pm 61.46$ | 385.42 | 0.18 |
| RS-100.0 | $118.50 \pm 62.54$ | 314.93 | 0.20 |

Table 1: Overall performance across domains, safety coefficients and thresholds.

D4PG and RC-D4PG in terms of penalized return. Figure 2 includes the reward shaping baselines. As can be seen in this figure, choosing a different reward shaping value can lead to drastically different performance. This is one of the drawbacks of the RS-D4PG variants. It is possible however, to find comparable RS variants (e.g., $RS - 0.1$ for the lowest safety coefficient of $0.05$). However, as can be seen in Figure 3, for the highest safety coefficient and largest threshold, this RS variant fails completely at the humanoid task, further highlighting the instability of the RS approach. Figure 3 which presents the performance of MetaL and the baselines on the highest safety coefficient and largest threshold (to ensure that the constraint task is solvable), shows that MetaL has comparable performance to RC-D4PG (a hard constrained optimization algorithm). This further highlights the power of MetaL whereby it can achieve comparable performance when the constraint task is solvable compared to hard constrained optimization algorithms and state-of-the-art performance when the constraint task is not solvable.

**Performance per domain:** When analyzing the performance per domain, averaging across safety coefficients and constraint thresholds, we found that MetaL has significantly better penalized return compared to D4PG and RC-D4PG across the domains. A table of the results can be seen in the Appendix, Figure 7. Note that, as mentioned previously, the RS-D4PG variants fluctuate drastically in performance across domains.

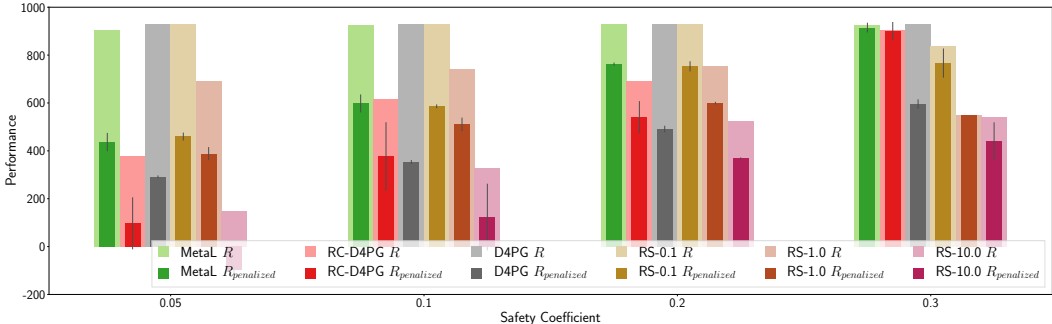

Figure 2: Performance as a function of safety coefficient.

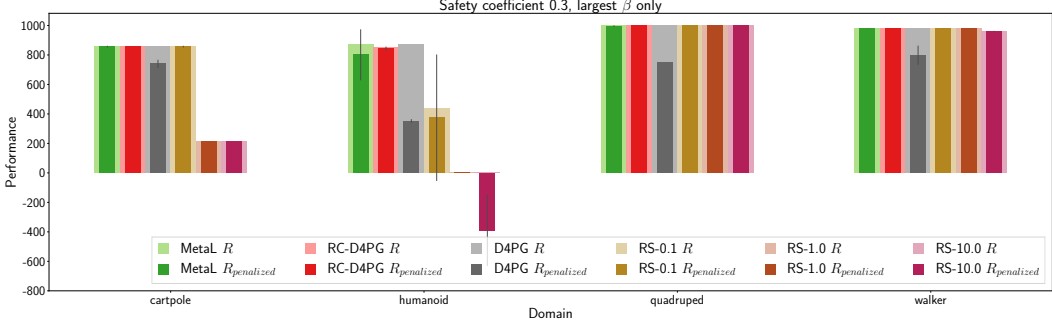

Figure 3: Performance per domain. MetaL compared to baselines in terms of average reward and penalized reward across the highest safety coefficient and largest thresholds for each domain.

**Algorithm behaviour analysis**: Since MetaL is a soft-constrained adaptation of RC-D4PG, we next analyze MetaL's gradient update in Theorem 1 to understand why the performance of MetaL differs

from that of RC-D4PG in two types of scenarios: (1) solvable and (2) unsolvable constraint tasks. For both scenarios, we investigate the performance on cartpole for a constraint threshold of $0.115$[6].

For (1), we set the safety coefficient to a value of $0.3$. The learning curve for this converged setting can be seen in Figure 4 (left). We track 4 different parameters here: the Lagrangian multiplier $\lambda$ (red curve), the mean penalty value $J_C^\pi$ (orange curve), the meta-parameter $\alpha_\lambda$ (black curve) and the scaled Lagrangian learning rate $\alpha_1 \cdot \exp(\alpha_\lambda)$ (green curve). The threshold $\beta$ is shown as the blue dotted line. Initially there are many constraint violations. This corresponds to a large difference for $J_C^\pi - \beta$ (orange curve minus blue dotted line) which appears in the gradient in Theorem 1. As a result, the meta-parameter $\alpha_\lambda$ increases in value as seen in the figure, and therefore increases the scaled learning rate to modify the value of $\lambda$ such that an improved solution can be found. Once $J_C^\pi$ is satisfying the constraint in expectation ($J_C^\pi - \beta \approx 0$), the scaled learning rate drops in value due to $J_C^\pi - \beta$ being small. This is an attempt by the algorithm to slow down the change in $\lambda$ since a reasonable solution has been found (see the return for MetaL (green curve) in Figure 4 (right)).

For (2), we set the safety coefficient to a value of $0.05$ making the constraint task unsolvable in this domain. The learning curves can be seen in Figure 4 (middle). Even though the constraint task is unsolvable, MetaL still manages to yield a reasonable expected return as seen in Figure 4 (right). This is compared to RC-D4PG that overfits to satisfying the constraint and, in doing so, results in poor average reward performance. This can be seen in Figure 4 (middle) where RC-D4PG has lower overshoot than MetaL for low safety coefficients. However, this is at the expense of poor expected return and penalized return performance as seen in Figure 4 (left). We will now provide some intuition for MetaL performance and relate it to the $\alpha_\lambda$ gradient update.

In this setting, there are consistent constraint violations leading to a large value for $J_C^\pi - \beta$. At this point an interesting effect occurs. The value of $\alpha_\lambda$ decreases, as seen in the figure, while it tries to adapt the value of $\lambda$ to satisfy the constraint. However, as seen in the gradient update, there is an exponential term $\exp(\alpha_\lambda)$ which scales the Lagrange multiplier learning rate. This quickly drives the gradient down to $0$, and consequently the scaled Lagrange multiplier learning rate too, as seen in Figure 4 (middle). This causes $\lambda$ to settle on a value as seen in the figure. At this point the algorithm optimizes for a stable fixed $\lambda$ and as a result finds the best trade-off for expected return at this $\lambda$ value. In summary, MetaL will maximize the expected return for an 'almost' fixed $\lambda$, whereas RC-D4PG will attempt to overfit to satisfying the constraint resulting in a poor overall solution.

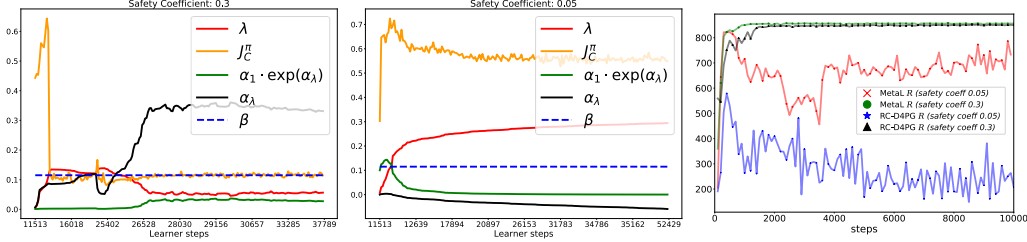

Figure 4: The learning progress of MetaL for solvable (left) and unsolvable (middle) constraint tasks. In both cases, MetaL attempts to try and maximize the return (right).

## 6 DISCUSSION

In this paper, we presented a soft-constrained RL technique called MetaL that combines meta-gradients and constrained RL to find a good trade-off between minimizing constraint violations and maximizing returns. This approach (1) matches the return and constraint performance of a hard-constrained optimization algorithm (RC-D4PG) on "solvable constraint tasks"; and (2) obtains an improved trade-off between maximizing return and minimizing constraint overshoot on "unsolvable constraint tasks" compared to the baselines. (This includes a hard-constrained RL algorithm where the return simply collapses in such a case). MetaL achieves this by adapting the learning rate for the Lagrange multiplier update. This acts as a proxy for adapting the lagrangian multiplier. By amplifying/dampening the gradient updates to the lagrangian during training, the agent is able to influence the tradeoff between maximizing return and satisfying the constraints to yield the behavior of (1) and (2). We also implemented a meta-gradient approach called MeSh that scales and offsets the

---

[6]This threshold was chosen as varying the safety coefficient at this threshold yields both solvable and unsolvable constraint tasks which is important for our analysis.

shaped rewards. This approach did not outperform MetaL but is a direction of future work. The algorithm, derived meta-gradient update and a comparison to MetaL can be found in the Appendix, Section B. We show that across safety coefficients, domains and constraint thresholds, MetaL outperforms all of the baseline algorithms. We also derive the meta-gradient updates for MetaL and perform an investigative study where we provide empirical intuition for the derived gradient update that helps explain this meta-gradient variant's performance. We believe the proposed techniques will generalize to other policy gradient algorithms but leave this for future work.

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
