# OpenReview forum: "Balancing Constraints and Rewards with Meta-Gradient D4PG"
_ICLR.cc/2021/Conference — ICLR 2021 Poster_

### Official Review · AnonReviewer2 · 2020-10-21
**Proposed methods is novel, evaluation is not considerably convincing (edit)**

**Rating:** 6
**Confidence:** 4

**Review:**

This paper addresses the soft constraints problem in RL. The problem is formulated as a Lagrangian optimization following Tesslaer et al. (2018) where the constraint is treated as a penalty in the reward. The base solution to the Lagrangian optimization is D4PG.
To adapt the learning rate of the Lagrangian multiplier and find a good trade-off between reward and penalty, the paper customizes the usage of meta-gradient method (Xu et al., 2018) to the problem in this paper. Two variants, MetaL and MeSH are designed.

The problem being addressed is an important topic in RL, the method proposed seems to be novel (but the intuition behind it is not clear), while the empirical evaluation is not convincing to me.

Strengths:
1) Addressing an important problem in RL, potentially could draw attention to the RL community
2) The paper provides a good motivation of addressing soft constraints in RL
3) Proposed method is somewhat novel

Weaknesses:
1) In the paper the motivation of using meta-gradient to solve the formulated Lagrangian optimization is only explained once at the beginning of Page 4 "Our intuition is that a learning rate gradient that takes into account the overall task objective and constraint thresholds will lead to improved overall performance." However it is clear to me what explicitly do you want to achieve? Are you trying to find the "ground truth" $\bar{\lambda}$ (i.e., 1000 in the experiments)? Does not seem to be the case; Do you want to somehow learn a "robust" policy that works well for all $\bar{\lambda}$ values? If so it is expected that the authors would show the empirical results on different $\bar{\lambda}$ values, and the proposed method works well in all of them.
2) Following 1), the paper evaluates different forms of outer loss and show that using a critic-only outer loss yields the best performance. What is the intuition behind this? It would be good that the authors can have a more in-depth discussion on this.
3) In the empirical evaluations, the paper defines a "penalized return". This is very un-intuitive -- in reality there is not always such quantized trade-off between reward and penalty of constraint violation. If there is, then one could directly add that to the objective. It would be more interesting to see, for example, given the same reward value, the proposed method always outperforms the baselines in penalty; or the other way round. The "penalized return" metric is therefore un-convincing to me.
4) Assuming the penalized return metric makes sense. From Figure 3 it appears that the performances of the baselines sometimes are very close to the proposed method. What puzzles me is that the performances of the baselines vary dramatically across domains. Can the authors elaborate more on why this happens?

Questions: See weaknesses points 1) 2) and 4)

Less important points:
1) In table 1 it seems that overall performance of RS-D4PG monotonically increases w.r.t. $\lambda$ values. I am curious to see what happens when $\lambda$ is even smaller.
2) Page 3, Line 2, $J_{obj}^{\pi}(\theta)$ -> $\tau$ and $\eta$ are missing in the bracket
3) Line 4 at paragraph D4PG: $Q_T(s, ...)$ -> s'

---

> ### Author Response · Authors · 2020-11-17
> **Thank you for your in-depth feedback. We hope we have addressed your concerns.**
>
> Thank you taking the time to do the review. We really appreciate your feedback and hope our response addresses your concerns.
>
> “Are you trying to find the "ground truth" [lambda]?”
>
> Thank you for the question. \bar{lambda} is actually a fixed *evaluation* weight that helps define the penalized return metric which is used to rank the performance of the converged policies. The name \bar{lambda} is probably confusing and we have changed the name of this constant in the updated paper.
>
> The value of \bar{lambda} is specifically chosen to be 1000 to ensure that both return (which has a maximum value of 1000 across all domains) and constraint overshoot (which is between 0 and 1) are considered equally when ranking policies.
>
> " However it is [un]clear to me what explicitly do you want to achieve?”
>
>  We explicitly want to achieve a soft-constrained RL optimization algorithm that (1) matches the return and constraint performance of a hard-constrained optimization algorithm (RC-D4PG based on [1]) on "solvable constraint tasks"; and (2) obtains an improved trade-off between maximizing return and minimizing constraint overshoot on "unsolvable constraint tasks" (This is compared to hard-constrained RL where the return simply collapses in such a case).
>
> We achieve this by adapting the Lagrangian multiplier learning rate using meta-gradients. This acts as a proxy for adapting the Lagrangian multiplier (i.e., lambda). By amplifying/dampening the gradient updates to the Lagrangian multiplier during training, the agent is able to influence the tradeoff between maximizing return and satisfying the constraints. As we show in the paper in the behaviour analysis section on page 8, this enables the agent to achieve (1) and (2) mentioned above. We have tried to make this clearer in the Discussion in our paper.
>
>
> “... intuition behind [critic outer loss]? … discussion on this.”
>
> Great question. The critic loss aims to learn a value function by minimizing the Temporal Difference (TD) error. In our algorithms, the critic loss is a function of lambda. As a result of this relationship, the value of lambda affects the agent's ability to minimize this loss and therefore learn an accurate value function. In D4PG, an accurate value function (i.e., the critic) is crucial for learning a good policy (i.e., the actor). This is because the policy relies on an accurate estimate of the value function to learn good actions that maximize the return (see D4PG actor loss). This would explain why adding the actor loss to the outer loss doesn’t have much effect on the final quality of the solution. We have updated the outer loss experiments section to capture this intuition.
> In addition, a number of meta-gradient papers (e.g., [2,3]) have found the critic loss to be empirically important in the outer loss.
>
>
> “..."penalized return" [is] un-intuitive...”
>
> There are certainly a variety of possible metric proposals (e.g., see [4]) as to how one can best capture the tradeoff between maximizing return and satisfying constraints. We wanted to capture performance in a single metric and, while we agree that this metric might not be suited to all problem domains, we found it to be particularly useful for comparing algorithms in our setup where all reward scales are the same and the penalties are between 0 and 1. Note that we also report the individual metrics of the average return and constraint overshoot (e.g., see Table 1) to provide readers with the full picture.
>
>
> “... Figure 3 ... baselines sometimes are very close to the proposed method.”
>
> Figure 3 shows the performance of all methods in a *solvable constraint task* setting: where we know that the task can be solved and the constraints can be satisfied. In this setup, the hard constrained RL algorithm RC-D4PG finds a constraint satisfying solution. In Figure 3 we actually *expect* the performance of MetaL to match the performance of RC-D4PG in this setting. As seen in the figure the reward shaping algorithms, which are known to be highly sensitive to the shaping parameter (lambda in this case) greatly fluctuate in performance, even in this “easy” setting.
>
> “... performances of the baselines vary dramatically across domains...”
>
> Good question. We assume the reviewer is referring to the Reward Shaping (fixed lambda) RL algorithms (RS-0.1, RS-1.0, RS-10, RS-100). Reward shaping in constrained RL is known to be highly sensitive to the value of lambda (e.g., see[1, 5]). In addition, these values need to be tuned for each individual task. As such, we expect performance to vary dramatically for each reward shaping baseline across the domains.
>
> * [1] Reward Constrained Policy Optimization, ICLR, 2018
> * [2] A self-tuning actor-critic algorithm, NeurIPS, 2020
> * [3] Meta-gradient reinforcement learning with an objective discovered online, NeurIPS, 2020
> * [4] Benchmarking Safe Exploration in Deep Reinforcement Learning, 2019
> * [5] Constrained Policy Optimization, ICML, 2017

---

> > ### Comment · AnonReviewer2 · 2020-11-22
> > **Author response clarifies most of my questions; still not fully convinced by the metrics used but willing to increase score**
> >
> > I did a re-read of the revised paper. Together with the extensive responses from the authors (really appreciate it), there are some points which I previously misunderstood (weaknesses points 1 and 4) that are now clarified. Also there are points that are better explained in the response (weakness point 2). I am not fully convinced by the metric being used, but I am willing to increase my score.

---

> > > ### Author Response · Authors · 2020-11-24
> > > **Thank you for your positive feedback**
> > >
> > > Thank you for taking the time to look over our response. We appreciate that you are willing to increase your score. Please let us know if there is anything else we can clarify.

---

### Official Review · AnonReviewer3 · 2020-10-28
**Simple approach to soft-constrained deep RL optimization with exhaustive, convincing evaluation**

**Rating:** 7
**Confidence:** 4

**Review:**

This paper proposes a simple approach to soft-constrained deep RL optimization using the unconstrained Lagrangian, by meta-learning the learning rate of the Lagrange multiplier. The authors include extensive evaluation, comparisons and ablations on 4 mujoco domains that establish the efficacy of the approach and provide insight into why it works better.

Pros:
1. Proposes a solution to the important problem of soft constraint optimization in deep RL, which is a challenge to be addressed for real world deployment. This work casts the unconstrained Lagrangian problem in the meta-learning framework, where the inner update adapts the learning rate for the Lagrange multiplier, and the outer loop optimizes the overall loss which takes into account both reward and constraint penalty. This adaptable learning rate prevents the optimizer from fitting too soon to the constraints and enables it to find a solution that can give higher overall performance (sum of reward and penalty).

2. The extensive evaluation includes comparison to prior approaches for constraint optimization in deep RL (soft-constraint with fixed Lagrangian multipliers and hard-constraint with learnable multipliers), tests with varying degree of constraint violations in the environment, ablations with variants of the outer loss across 4 mujoco domains in the Real-world RL suite. The authors also include an analysis of how the and Lagrange multiplier and its learning rate evolve in a setting with high number of safety violations, to show the benefit of using the adaptable learning rate.


Cons :
1. The idea of adapting the learning rate in the inner loop of a meta-learning algorithm is not novel, and is often used to enhance performance. With that said, this work convincingly shows that using this simple idea is effective for constrained optimization in the meta-learning framework.

2. The paper introduces another variant of their main approach, but doesn't discuss it adequately in the main paper, and it's unclear why it does worse than the proposed approach.

3. The paper could be further strengthened by experiments on real world domains, where soft-constrained optimization is a critical challenge.

---

> ### Author Response · Authors · 2020-11-17
> **Thank you for your positive feedback.**
>
> Thank you for taking the time to do the review. We really appreciate your helpful feedback.
>
> “The idea of adapting the learning rate in the inner loop of a meta-learning algorithm is not novel, and is often used to enhance performance. With that said, this work convincingly shows that using this simple idea is effective for constrained optimization in the meta-learning framework.”
>
> We appreciate your feedback. We actually hope that, since this is a simple idea, it can be easily adapted into other constrained RL variants. In addition, there are other parameters that can also be optimized using meta-gradients such as the learning rates of the actor and critic for example, which we leave for future work.
>
> “The paper introduces another variant of their main approach, but doesn't discuss it adequately in the main paper, and it's unclear why it does worse than the proposed approach.”
>
> We have moved MeSH to the Appendix as we agree that it is not adequately discussed in the main paper. In the Appendix, you can find a thorough analysis of MeSH which includes a derivation of the meta-gradient update as well as an in-depth analysis of its performance. We briefly mention MeSH in the discussion.
>
> “The paper could be further strengthened by experiments on real world domains, where soft-constrained optimization is a critical challenge.”
>
> Having experiments on real-world domains would definitely strengthen the paper. We are actively working on running this approach on real-world domains and hope to add this to a follow-up work.

---

### Official Review · AnonReviewer4 · 2020-10-28
**well written paper - could have a large impact**

**Rating:** 7
**Confidence:** 4

**Review:**

The paper presents two soft-constrained rl approaches built on top of D4PG. Specifically, they use meta gradients for the lagrange multiplier learning rate (MetaL), and use meta gradients for reward shaping (MeSH).

I found the paper to very clearly written, the main algorithm MetaL is clearly presented, and the results are fairly conclusive: the meta gradient approach proposed in the paper works better than the tested baselines. The introduction motivates the different real-world problems very nicely, e.g., hard vs. soft constraints. As someone with a lot of deep RL experience, but not a lot of constrained RL experience, I found the authors did a very good job at explaining all the relevant background. I also really appreciated the detailed experimental analysis of the approach at the end of 6.1 – it highlights exactly why the method works well.

One critique I do have, is that it would be great to have the intuition for the MeSH update in the main paper. Otherwise, since the results indicate that it performs worse than MetaL across the board – perhaps it would be best to relegate the method to the appendix and change the presentation of the paper, e.g., change “we propose two meta gradient methods”  etc… I feel like the paper would be stronger without MeSH as it takes away from the overall message.

Another issue I have is that 3 seeds might not be enough to make any meaningful conclusions. More seeds would be needed to make any statistical comparison between metal and Rs-0.1 in table 1. Nevertheless, as highlighted RS-0.1 fails at humanoid – which makes sense – you would need to tune the penalty parameter for each domain (although 0.1 actually works quite well on ¾ domains). Which is why meta gradient approaches make sense.

Overall, assuming the authors add extra seeds and perform the statistical significance testing – I think this paper would have a large impact at ICLR.

Small notes:

On the presentation side of the results, I find figures 2 and 3 to be hard to interpret at a glance. It would be better to compare against the best baseline – instead of ~6 of them.

One final note, \bar \lambda = 1000 corresponds to the upper bound of the reward. Unless the algos are consistently achieving this upper bound, \bar \lambda seems very high. Ideally, a bunch of lambdas should be tested. This feels arbitrary.

---

> ### Author Response · Authors · 2020-11-17
> **Thank you for the positive feedback.**
>
> Thank you for taking the time to do the review. We really appreciate your helpful feedback.
>
> “perhaps it would be best to relegate the method to the appendix and change the presentation of the paper, e.g., change “we propose two meta gradient methods” etc… I feel like the paper would be stronger without MeSH as it takes away from the overall message.”
>
> Good point. We agree that relegating MeSH to the appendix would make the paper story clearer and not detract from the main message. We have made these changes in the latest revision.
>
>
> “Another issue I have is that 3 seeds might not be enough to make any meaningful conclusions. More seeds would be needed to make any statistical comparison between metal and Rs-0.1 in table 1. Nevertheless, as highlighted RS-0.1 fails at humanoid – which makes sense – you would need to tune the penalty parameter for each domain (although 0.1 actually works quite well on ¾ domains). Which is why meta gradient approaches make sense.”
> “Overall, assuming the authors add extra seeds and perform the statistical significance testing – I think this paper would have a large impact at ICLR.”
>
> We appreciate your positive feedback. We have added 5 additional seeds (8 in total) and have updated the plots and tables in the paper. We used Welch’s t-test to test the significance and found that MetaL has significantly better performance than the baselines as all p-values are less than 1e-09. We have reported this in the paper as a footnote.
>
> “On the presentation side of the results, I find figures 2 and 3 to be hard to interpret at a glance. It would be better to compare against the best baseline – instead of ~6 of them.”
>
> Good point. We have added a figure comparing MetaL vs. RC-D4PG vs. D4PG (i.e. soft vs. hard constrained optimization vs. ignoring constraints). We still want to keep Figures 2 and 3 as these figures showcase the instabilities of the reward shaping variants. However, we hope that the additional figures you requested make the results clearer to interpret. If you would like any other changes, please let us know.
>
> “One final note, \bar \lambda = 1000 corresponds to the upper bound of the reward. Unless the algos are consistently achieving this upper bound, \bar \lambda seems very high. Ideally, a bunch of lambdas should be tested. This feels arbitrary.”
>
> We did indeed try a variety of lambdas, and found that, for our particular domains and reward/penalty functions, this value nicely captures the trade-off between maximizing return and minimizing constraint violations when computing penalized reward. We agree that this metric is not perfect, since a different \bar \lambda may be better suited to other domains. As such, we also report the individual metrics to give readers a full overview of our results.

---

### Official Review · AnonReviewer1 · 2020-10-29
**Recommendation to Accept**

**Rating:** 7
**Confidence:** 4

**Review:**

#### Summary:
The paper focuses on soft-constrained RL techniques and proposes a meta-gradient approach for the same. It first extends the RCPO (Tessler et al)  algorithm using the methodology of DDPG (Lillicarp et al) to propose an off-policy version of RCPO (called RC-D4PG). The main contribution of the work is the proposal of two new meta-gradients based algorithms for the soft-constrained RL problem that are able to find a good trade-off between constraint violation and maximizing returns. The first proposed algorithm - Meta-L - is based on a meta-learning based adaptive update rule for the Lagrange multiplier's learning rate. The second algorithm is based on similar principles but instead focuses on adapting the reward-shaping update in a meta manner. The author's show the empirical evidence of their method's strengths on a bunch of continuous control based simulator tasks.

#### Strengths:

- **Setting**: The paper shifts focus on the soft-constrained RL (a relaxed CMDP) setting and provide good motivation to work on these setting rather than sticking to the much harder CMDP setting. This makes sense especially for the choice of problems the authors are interested in such as saving energy costs, find some trade-off between two metrics, etc.

- **Method**: One of the biggest problems with the Lagrange Optimization-based CMDP algorithms is that the optimization of the Lagrange multiplier is tricky (and most times very brittle). I like the approach of the authors that aims to adaptively modify the rate of change of the Lagrange multiplier, and I believe that it is a step forward in the right direction.

- **Empirical results**: The authors have strong empirical results for their methods, and it seems that the meta-gradient based approach is able to find the trade-off for the more successfully when compared to the baselines.

- **Analysis**: The empirical analysis of the behavior of the proposed algorithms has been insightful as a reader. I appreciate the extensive experiments and particularly enjoyed the section Algorithm behavior analysis.


#### Weakness:

- **Reproducibitlity**: There is no mention of code release. Though, they provide the architectural details in the appendix it is difficult to take the author's claim at face-value when the implementations of the Deep-RL-based methods aren't released.

- **Outer-losses**: The authors propose and test a variety of outer-loss and pick the best one here. I can see why the authors chose this approach, however, from an outsider's perspective, this means for any new application where this algorithm needs to be deployed, a lot of effort and computing needs to go first in finding the appropriate outer loss (on top of hyper-parameter optimization!). It'll be great to have some insights into the choice of outer-loss, other than the hyper-parameter based search that is currently being used.

- **Solution quality**: No comments have ever been mentioned about the quality of the solution reached. It'll be nice to have some guarantees associated with the solution quality in this case as the setting is motivated by safety and constraint violation.


#### Questions for rebuttal:

- Why is the Meta-parameter update for MetaL (in Algorithm 1, Line 17) different from the derived gradient update in Theorem 1 ?
- Does updating the Lagrange multiplier adaptively makes the initial learning rate selection process more cumbersome or fragile?
- Why did the authors choose the off-policy extension of RCPO to build upon, when the original RCPO was purely online? I'm curious to know what breaks (if at all anything breaks) with the original online formulation.

---

> ### Author Response · Authors · 2020-11-17
> **Thank you for your positive feedback.**
>
> Thank you for taking the time to do the review. We really appreciate your helpful feedback.
>
> “Reproducibility: … ”
>
> We do intend to release the code. We have implemented our methods using the Acme RL framework (https://github.com/deepmind/acme) and are actively working on open-sourcing our code.
>
> “Outer-losses: … It'll be great to have some insights into the choice of outer-loss....”
>
> Great question. The critic loss aims to learn a value function by minimizing the Temporal Difference (TD) error. In our algorithms, the critic loss is a function of lambda. As a result of this relationship, the value of lambda affects the agent's ability to minimize this loss and therefore learn an accurate value function. In D4PG, an accurate value function (i.e., the critic) is crucial for learning a good policy (i.e., the actor). This is because the policy relies on an accurate estimate of the value function to learn good actions that maximize the return (see D4PG actor loss). This would explain why adding the actor loss to the outer loss doesn’t have much effect on the final quality of the solution. We have updated the outer loss experiments section to capture this intuition.
> In addition, a number of meta-gradient papers (e.g., [1, 2]) have found the critic loss to be empirically important in the outer loss.
>
> “Solution quality: No comments have ever been mentioned about the quality of the solution reached. It'll be nice to have some guarantees associated with the solution quality in this case as the setting is motivated by safety and constraint violation.”
>
> Great question. We can guarantee that our algorithm converges to a locally optimal solution and are happy to add this to the Appendix if you feel this is a necessary addition. However, this does not explicitly capture solution quality with respect to safety. This is an interesting direction and one that we are actively pursuing for future work.
>
> “Why is the Meta-parameter update for MetaL (in Algorithm 1, Line 17) different from the derived gradient update in Theorem 1?”
>
> They are the same. The meta-parameter update for MetaL in the algorithm is the shortened form of the gradient update in Theorem 1. We have made this connection more explicit in the paper.
>
> “Does updating the Lagrange multiplier adaptively make the initial learning rate selection process more cumbersome or fragile?”
>
> Great question. We actually ran an experiment to compare the performance of MetaL at different initial learning rates to RC-D4PG (which fixes the learning rate throughout training). We found that both methods are sensitive to the initial learning rate. However, MetaL consistently yields significantly higher penalized return performance compared to RC-D4PG across all the tested learning rates. We have included the plot in the Appendix as Figure 7.
>
> “Why did the authors choose the off-policy extension of RCPO to build upon, when the original RCPO was purely online? I'm curious to know what breaks (if at all anything breaks) with the original online formulation.”
>
> Great question. RCPO was implemented on top of the online PPO algorithm (see [1]). D4PG outperforms PPO as shown in [2] (see Figure 5). As the reviewer mentioned, D4PG is off-policy and, as such, we've adapted RCPO to support off-policy learning (yielding RC-D4PG).
> That being said, we don’t expect anything to break in the online setting. Rather, we wanted to showcase the performance of our agent on a state-of-the-art continuous control RL algorithm.
>
> * [1] Reward Constrained Policy Optimization, ICLR, 2018
> * [2] Distributed Distributional Deterministic Policy Gradients (D4PG), 2018

---

### Decision · Program_Chairs · 2021-01-07
**Final Decision**

**Decision:**

Accept (Poster)

**Comment:**

The paper looks at the soft-constrained RL techniques and proposes a meta-gradient approach.
One of the biggest problems with the Lagrange Optimization-based CMDP algorithms is that the optimization of the Lagrange multiplier is tricky
The proposed solution and empirical results have promise. The reviewers broadly agree on their evaluation and the major concerns on comprehension, additional experiments and as well as comparison with baselines have been addressed in the rebuttal.

- Convergence rate and quality of fixed point reached.
The authors mention convergence to local optima but omit the quality of this solution from perspective of safety. It would be useful to include a discussion on the topic, with potential references to concurrent work.
Other relevant and concurrent papers to potentially take note of:
- Risk-Averse Offline Reinforcement Learning (https://openreview.net/forum?id=TBIzh9b5eaz)
- Distributional Reinforcement Learning for Risk-Sensitive Policies (https://openreview.net/forum?id=19drPzGV691)
- Conservative Safety Critics for Exploration (https://openreview.net/forum?id=iaO86DUuKi)

I would recommend acceptance of the paper based on empirical results, conditional on release of sufficiently documented and easy to use implementation.
Given the fact that the main argument is empirical utility of the method, it would be limit the impact of this work if readers cannot readily build on this method.